# Forest Structure and Projections of *Avicennia germinans* (L.) L. at Three Levels of Perturbation in a Southwestern Gulf of Mexico Mangrove

Agustín de Jesús Basáñez-Muñoz [1], Adán Guillermo Jordán-Garza [2,*] and Arturo Serrano [3]

[1] Facultad de Ciencias Biológicas y Agropecuarias, Cuerpo Académico Manejo de Ambientes Marinos y Costeros, Universidad Veracruzana, Tuxpan 92870, Mexico; abasanez@uv.mx

[2] Coral Reefs Laboratory, Facultad de Ciencias Biológicas y Agropecuarias, Cuerpo Académico Ecosistemas Costeros, Universidad Veracruzana, Tuxpan 92870, Mexico

[3] Observatorio Marino y Costero, Facultad de Ciencias Biológicas y Agropecuarias, Universidad Veracruzana, Tuxpan 92870, Mexico; arserrano@uv.mx

* Correspondence: ajordan@uv.mx; Tel.: +52-7831469318

**Abstract:** Mangrove forests have declined worldwide and understanding the key drivers of regeneration at different perturbation levels can help manage and preserve these critical ecosystems. For example, the Ramsar site # 1602, located at the Tampamachoco lagoon, Veracruz, México, consists of a dense forest of medium-sized trees composed of three mangrove species. Due to several human activities, including the construction of a power plant around the 1990s, an area of approximately 2.3 km$^2$ has suffered differential levels of perturbation: complete mortality, partial tree loss (divided into two sections: main and isolated patch), and apparently undisturbed sites. The number and size of trees, from seedlings to adults, were measured using transects and quadrats. With a matrix of the abundance of trees by size categories and species, an ordination (nMDS) showed three distinct groups corresponding to the degree of perturbation. Projection matrices based on the size structure of *Avicennia germinans* showed transition probabilities that varied according to perturbation levels. Lambda showed growing populations except on the zone that showed partial tree loss; a relatively high abundance of seedlings is not enough to ensure stable mangrove dynamics or start regeneration; and the survival of young trees and adult trees showed high sensitivity.

**Keywords:** mangrove; Gulf of México; population; Leslie matrix

## 1. Introduction

Mangrove forests are diverse ecosystems that provide human communities with many goods and services [1]. Unfortunately, recent climate change is altering the latitudinal limits of woody assemblages [2], and sea-level rise will likely provoke catastrophic changes in most wetlands due to inundation and salinity changes [3,4]. At a more regional and local level, coastal development causes land-cover changes due to different human activities [5–7]. Mangroves along the Gulf of Mexico have declined due to urban and harbor development, energy generation, and tourism development [8,9]. Understanding the main drivers of decline and recovery at small spatial scales can help decide on the best management practices to protect and rescue these highly valuable ecosystems [10]. In the coastal area of the southwestern Gulf of Mexico, along with three Mexican states (Tamaulipas, Veracruz, and Tabasco), a fringe of approximately 87,047 ha of mangrove forests is present [11]. Four mangrove species that vary in relative abundance dominate the structure of these forests: *Rhizophora mangle* L., *Avicennia germinans* (L.) L., *Laguncularia racemosa* (L.) Gaertn., and *Conocarpus erectus* L. [12]. It has been shown that well-preserved mangrove communities are composed, with local variations, of these mangrove species and that the enrichment of the floristic assemblages can, in fact, be a sign of perturbation [13]. In Veracruz, the mangrove fringe is not continuous and is divided into smaller patches (<1000 ha) located

between the coast and large coastal lagoons such as Tamiahua, Tampamachoco, and San Agustín [12]. Along the Tampamachoco lagoon, Ramsar site number 1602 was established by Mexico in 2006 due to its vast and well-preserved mangrove area, one of the largest in the Gulf of Mexico. Its structure, composition, productivity, phenology, restoration, and uses have been studied by various local universities—particularly an area of approximatively 40 ha, located near the power plant [14–18]. In the 1980s, the construction of three embankments, built to support power towers, changed the average water circulation, and 20 ha of mangrove developed signs of top dying [17]. This mortality continued until an area between the power plant and the Tampamachoco lagoon ultimately died. Although the activities to set up the power plant were probably the origin of the mortality, there is still no explanation of why the area has not recovered, why restoration efforts have failed, and mortality continues to expand. Within the affected area, five perturbation levels exist: complete mangrove mortality, partial mangrove mortality, apparently unaffected and isolated patches, and adjacent unaffected areas. The present study quantified mangrove community structure in these areas, highlighting structural differences according to perturbation levels. In addition, we used a matrix-population analysis of *A. germinans*, to assess the effects of transition probabilities and recruitment on the stability of the mangrove forest at the different perturbation areas.

## 2. Materials and Methods

The study site is located to the north of the Veracruz state, approximately 10 km to the west of Tuxpan, México (Figure 1).

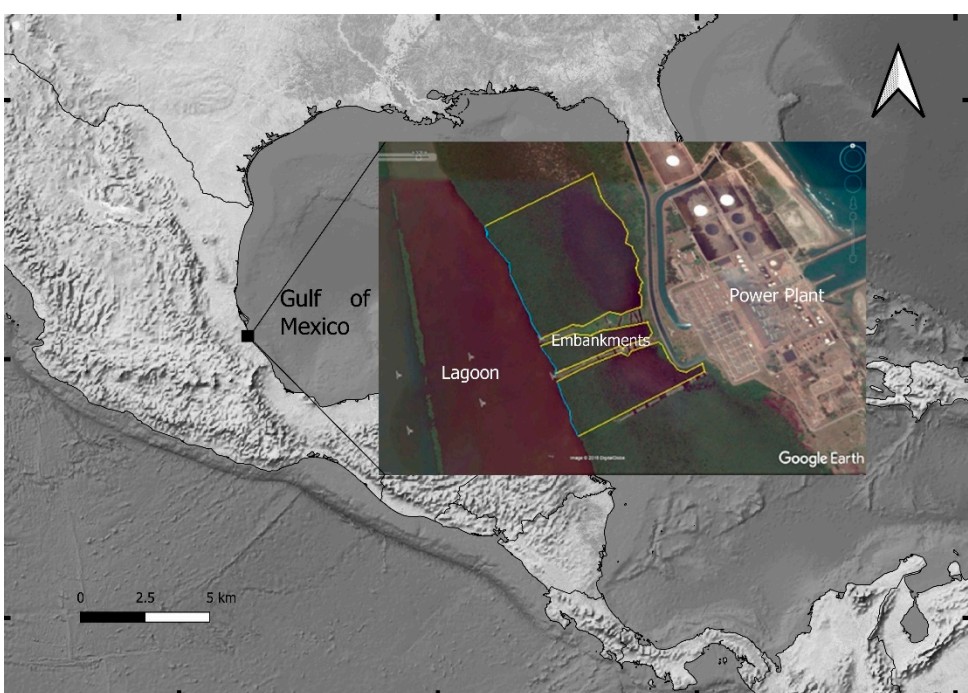

**Figure 1.** Map showing the location of the study at the northeast coast of the Gulf of Mexico. The mangrove is located between a powerplant to the east and the Tampamachoco lagoon to the west. The study area is marked in yellow and shows different levels of perturbation from the lagoon to the power plant: Apparently unaffected forest (AUF); Partial mangrove mortality (PMM); complete mangrove mortality (CMM); and a thin band of isolated apparently unaffected forest growing next to the power plant (AUF isolated). The embankments that support electric towers can be seen within the study area. Map made in QGIS 3.2 with Natural Earth, free vector and raster map data @ naturalearthdata.com (accessed on 24 July 2021). Satellite image from the study site in 2019 taken from Google Earth Pro.

Tuxpan is in the northern part of Veracruz, has minimum temperatures higher than 0 °C, and annual mean precipitation > 1750 mm, conditions that are suitable for these tree species, common along the Gulf of Mexico coast [19]. The community structure at the mangrove located between the coastline to the east and the Tampamachoco lagoon to the west is given by three mangrove species (*R. mangle*, *A. germinans*, and *L. racemosa*) [16]. In 1985, the construction of a thermoelectric power plant began over the mangrove along the coastline (Figure 1). The establishment of the power plant divided the remaining mangrove, towards the lagoon, into small sections due to the construction of three embankments where lattice towers, supporting power lines, were built (Figure 1). The first section to the north (I) has an area of approximatively 90.7 ha, the second section at the center (II) has an area of around 5.5 ha, and the last section, to the north (III), is the largest with approximatively 15.5 ha. Within each of these sections, four levels of perturbation can be found from the powerplant towards the lagoon: an isolated patch that remained in an elevated terrain adjacent to the powerplant (ISO); complete mangrove mortality (CMM); partial mangrove mortality (PMM); and an apparently unaffected forest (AUF, Figure 1). The forest adjacent to the affected area was used as a control section (Figure 1).

A total of 5 transects approximately 500 m long were laid, perpendicular to the coast and between the power plant and the lagoon. Two transects were in an adjacent, well-preserved area and used as controls, and three transects intersected all three levels of perturbation within the sections made by the embankments (Figure 2).

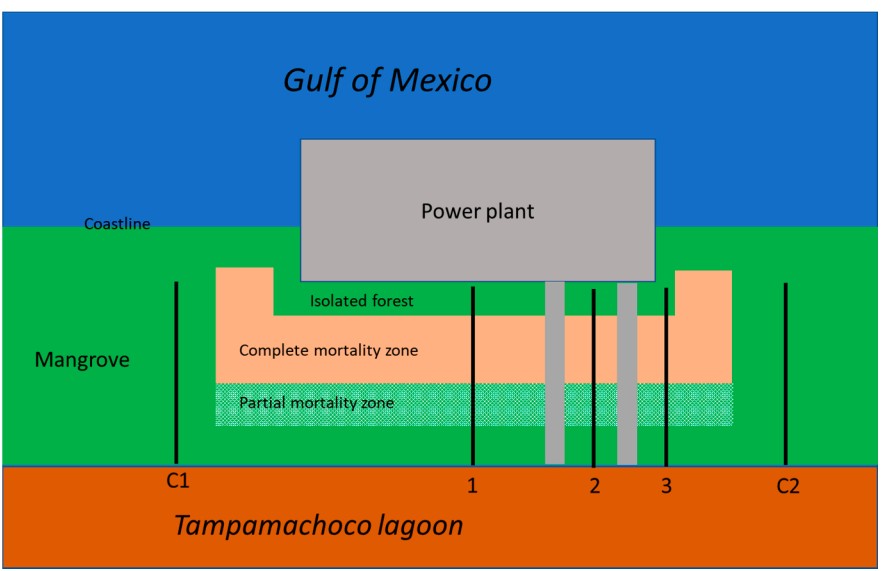

**Figure 2.** Schematic showing the sampling design. Three transects were laid within the study area and intercepted zones that included isolated apparently unaffected forest, complete mangrove mortality, partial mangrove mortality, and apparently unaffected forest. Two additional transects were laid outside the study area to the north and south as controls (C1 and C2, respectively).

Over every transect, a total of 4, 10 × 10 m quadrats were laid (i.e., 20, 10 × 10 quadrats). These quadrats were positioned on different perturbation zones defined a priori: (1) Apparently unaffected forest (AUF); (2) Partial mangrove mortality (PMM); (3) Complete mangrove mortality (CMM); and (4) Isolated apparently unaffected forest (isoAUF) (Figure 2). Within each large quadrat, a set of haphazardly laid quadrats were deployed: medium quadrats (5 × 5 m) and small quadrats (1 × 1 m). Individual trees were counted on each quadrat and classed into four size categories. The smallest categories were counted in the smallest quadrats, juveniles in the medium quadrats and adult trees in the largest [20,21]. Although the quadrats within the CMM zone were laid and visited, live trees were only found on the corresponding areas of the control transects (Figure 3).

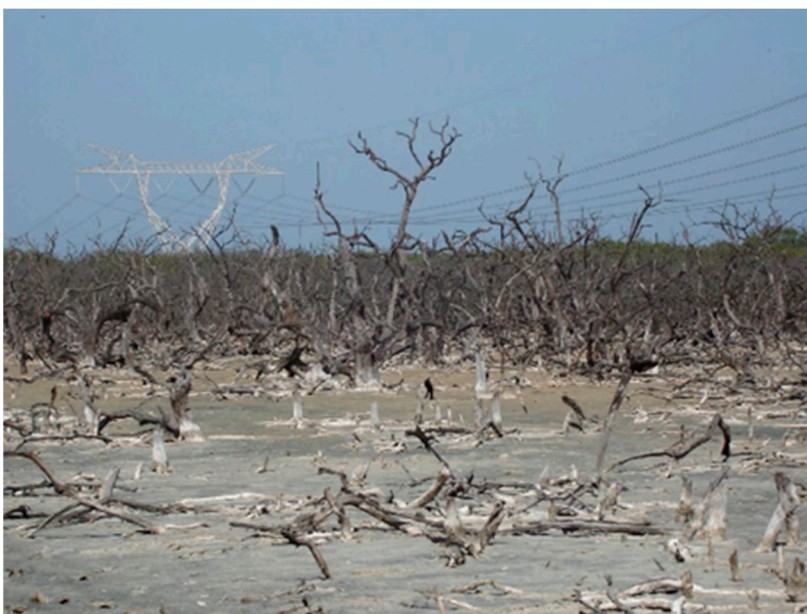

**Figure 3.** A view of the complete mangrove mortality area: no living trees at any developing stages, and only remains of old trees in a swamp were found.

The height of the trees from the ground to the top of the canopy and diameter at breast height (approximately 130 cm from the ground and 30 cm from the highest root in *R. mangle*) were measured [22,23] and used to classify the trees into four size classes. The size classes were defined as: <40 cm in height (seedlings); from 40 to <150 cm in height (juveniles); >150 cm in height but with <2.5 cm in diameter at breast height (DBH) (young tree); >150 cm in height and DBH between 2.5 to >10 cm (adult trees). These categories are based on *A. germinans*, for which the size when trees reach a reproductive age was considered a young tree and larger [24]. Because the sampled area of the different quadrats differed, the abundance observations were transformed to density (# of trees/m$^2$). Later, to construct size structures of *A. germinans*, the number of trees present of an area of 100 m$^2$ was estimated from the observed densities.

To examine potential differences in the forest structure, non-metric multi-dimensional scaling (nMDS) analysis was performed in PAST [25]. The data matrix consisted of the mean density of each tree species at their four size categories at each perturbation area by transect. A Bray–Curtis similarity matrix was calculated from the original density matrix to be used in the nMDS analysis. The Bray–Curtis index (BC$_{jk}$) gives the similarity between the *j*th and *k*th samples according to the following formula:

$$\text{BC}_{jk} = 100 \left[ 1 - \frac{\sum_{i=1}^{p} |Yij - Yik|}{\sum_{i=1}^{p} |Yij + Yik|} \right]$$

with *Yij* representing the mean density for the *i*th species/size class in the *j*th sample; and *Yik* being the mean density for the *k*th sample. The absolute value of these operations is taken and subtracted to 1; to have the index as a %, it is multiplied by 100 [26]. The nMDS analysis uses the Bray–Curtis similarity matrix and constructs a configuration of the samples (in 2 dimensions) that attempts to satisfy the conditions given by the similarity matrix; a stress value is calculated to assess the degree of distortion of this two-dimensional ordination; in general, a stress value < 0.2 indicates a useful ordination [26].

An analysis of similarities (ANOSIM) was used to conduct multiple comparisons between perturbation levels (factor with four levels: Control, apparently unaffected forest (AUF), isolated AUF and partial mangrove mortality). Finally, a percent similarities analysis (SIMPER) was used to determine which tree size and species contributed most to differences found [27,28].

The ANOSIM uses the rank similarities of the Bray–Curtis similarity matrix and calculates a coefficient R, that is defined as:

$$R = \frac{(rb - rw)}{0.5(M)}$$

where $rb$ is the average of rank similarities from all pairs of replicates between the factor levels and $rw$ is the average of rank similarities of replicates within the factor levels, $M$ is defined as $\frac{n(n-1)}{2}$ and $n$ is the total number of samples. R varies between $-1$ and 1 and it is close to 0 if the null hypothesis is true and close to 1 if replicates within a factor level are more similar than replicates from other factor levels [26]. The SIMPER analysis uses the Bray–Curtis similarity matrix and the same factor as ANOSIM; the average dissimilarity between all pairs of inter-factor samples is computed and used to assess the contribution of each species/size category to that average dissimilarity [26].

We used the species *A. germinans* to build size structures according to perturbation levels (control, AUF, AUF isolated, and PMM) because more accurate population information can be found for this species [20,29]. Derived from the observed densities, the number of trees in a 100 m$^2$ area was estimated for each size category (I, II, III, and IV). Transition probabilities were estimated as the rate of the number of trees in size category $i + 1$ divided by the number of trees in size category $i$. If the transition probability was larger than 1, it was set to 1, but a loop of 0.8 was established on the $i + 1$ category because more trees than expected were found. This happened mostly for trees in the larger size category, given that old surviving trees accumulate. Only trees on size categories III and IV were assumed reproductive [24], and the number of seedlings was adjusted to match the ones observed in size category I. According to the forest state, we used matrix-population projections: partial mangrove mortality (PMM), apparently unaffected forest (AUF), and AUF isolated and compared them to the control. The package "popbio" [30] was used in R [31]. For each projection, the initial population consisted of 1000 seedlings (size class I), and the trajectories of each population were simulated for 10 time steps. The projection matrix and initial state vector for each forest state followed the general form [32]:

$$\text{ProjectionsMatrices}: \begin{bmatrix} 0 & 0 & F3 & F4 \\ P1 & 0 & 0 & 0 \\ 0 & P2 & 0 & 0 \\ 0 & 0 & P3 & B1 \end{bmatrix} \times \begin{bmatrix} 1000 \\ 0 \\ 0 \\ 0 \end{bmatrix}$$

In the projection matrices, we define: $P1$ to $P3$ as the probability to pass from one size class to the next; $B1$ as the probability to stay on the same size class; and $F3$ and $F4$ as the fecundities for reproductive trees.

The first eigenvalue lambda ($\lambda$) was used to assess the growth of the population ($\lambda < 1$: population declines, $\lambda = 1$; stable population and $\lambda > 1$: population grows). In addition, a sensitivity analysis showed which elements of the matrices would have larger effects on lambda if their values changed [32].

## 3. Results

### 3.1. Forest Structure

Overall mean tree density was $0.2 \pm 0.8$ trees/m$^2$ and mean height was $185 \pm 181$ cm for trees larger than 40 cm in height of the three mangrove species found on the area: *A. germinans*, *L. racemosa*, and *R. mangle*. *L. racemosa* was the least abundant overall, while *R. mangle* was the most abundant species in the controls, followed by *A. germinans*, which reached larger sizes. Although density and height varied according to perturbation levels, all three species were absent in the most affected area, and on the partial mortality area, *L. racemosa* was absent, and *A. germinans* was the most abundant species (Figure 4).

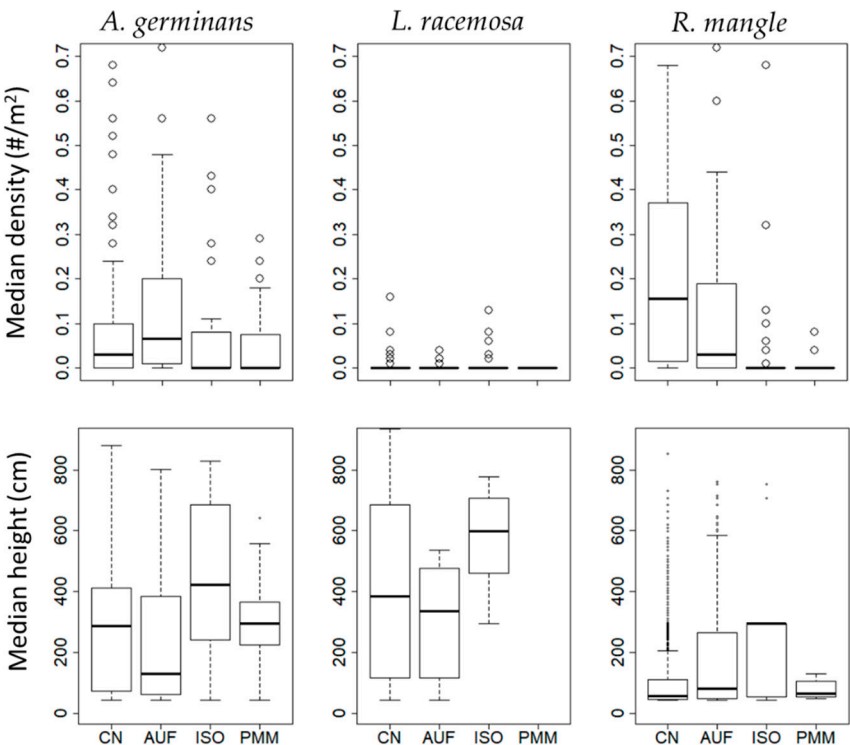

**Figure 4.** Boxplots showing density and tree height (not including size class I) of *Avicennia germinans* (L.) L., *Laguncularia racemosa* (L.) Gaertn. and *Rhizophora mangle* L. at different perturbation levels (CN = Control, AUF = Apparently unaffected forest, ISO = Isolates apparently unaffected forest, and PMM = Partial mortality mangrove).

Mean seedling density was $21.2 \pm 60.3$ seedlings/m$^2$, and this also varied by perturbation levels; seedlings in the partial mangrove mortality zone were particularly depleted compared to other zones (Figure 5).

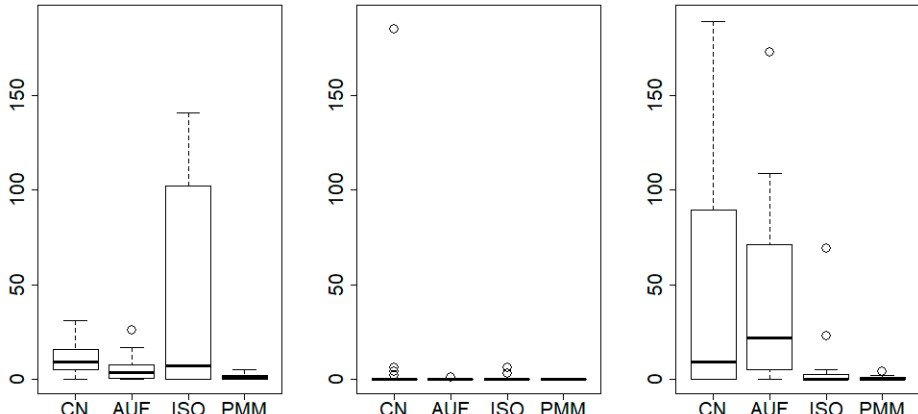

**Figure 5.** Boxplots showing density of seedlings (size class I) of *Avicennia germinans* (L.) L., *Laguncularia racemosa* (L.) Gaertn. and *Rhizophora mangle* L. at different perturbation levels (CN = Control, AUF = Apparently unaffected forest, ISO = Isolates apparently unaffected forest, and PMM = Partial mortality mangrove).

The forest structure differed between the three levels of perturbation and the controls (ANOSIM R: 0.5, *p* = 0.001; Figure 6). The nMDS shows that the samples on the PMM zone differ from the rest, and the isolated apparently unaffected forest seems to form a 3rd group (Figure 6).

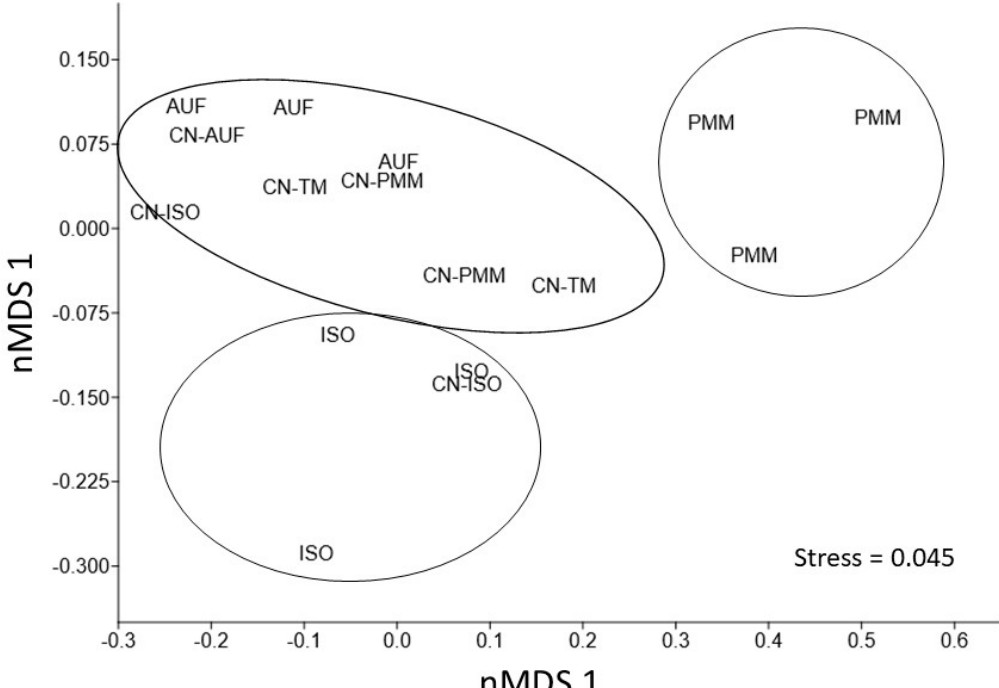

**Figure 6.** Ordination spaces showing sampled quadrats by perturbation level. (1) Apparently unaffected forest (AUF); (2) Partial mangrove mortality (PMM); and (4) Isolated apparently unaffected forest (ISO). Control quadrats are depicted by CN, in addition CN-AUF corresponds to a control quadrat that is located at a similar distance from the lagoon to the power plant as the AUF area, etc. Three main groups are apparent: the PMM quadrats (upper-right), the ISO quadrats (lower-left) and control quadrats mixed with AUF (upper-left).

Multiple comparisons, with a Bonferroni correction [33], showed significant differences only between the control and the partial mangrove mortality zones (R: 0.67, $p = 0.0067$). According to the SIMPER analysis, *R. mangle* seedlings (size class I) (53.7% contribution) and *A. germinans* seedlings (41.36% contribution) contributed 95.1% to the dissimilarity between the control and PMM zones, with both species having more seedlings on the controls.

### 3.2. Projections of Avicennia germinans (L.) L.

In all mangrove forest states, *A. germinans* showed size structures skewed to the right (Figure 7). These structures show large mortality from seedlings to size classes II and III, yet once trees reach larger sizes, their probability of survival increases (Figure 7).

The populations on each level of perturbation were projected for ten-time steps. The projections showed that the control, apparently unaffected forest (AUF) and isolated AUF increased through time ($\lambda = 1.21$; 1.21 and 1.1, respectively). For the control, transitions from class I to II (4.7) and from class II to III (0.98) were the most sensitive; for AUF, the most sensitive transition was from size class II to III (4.5) and from size class I to II (1.2), and for isolated AUF transitions, from class I to II (74) and from class II to III (0.5) were the most sensitive (Figure 8). On the other hand, the population at the PMM perturbation level showed a rapid decline ($\lambda = 0.8$), and although some seedlings reached size class II, these rapidly died (Figure 8). The highest sensitivity was related to the survival of adult trees (transition probability from size class IV to itself); if adult trees were present, then the population would be able to maintain itself if these adults survived and reproduced locally, yet the 100% from class II to class III causes a depletion of larger trees and the collapse of the population.

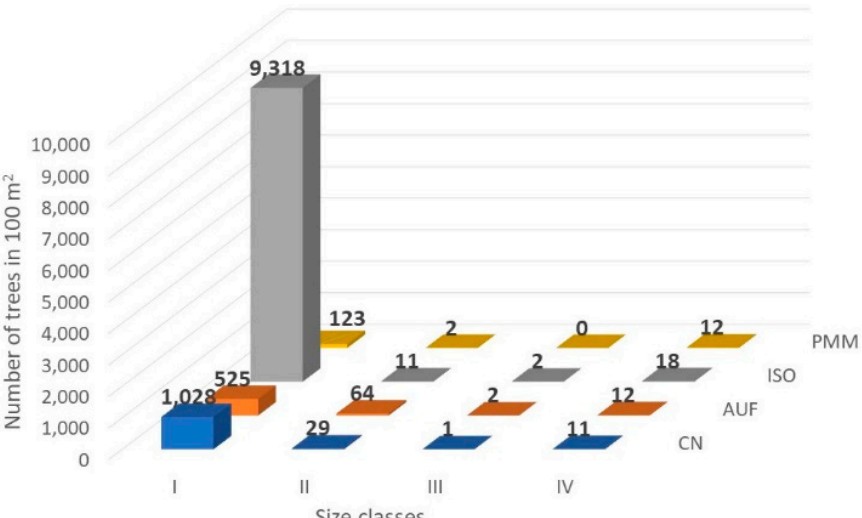

**Figure 7.** Size structures of *A. germinans* by perturbation zone (Control CN; Apparently unaffected forest AUF; Isolated apparently unaffected forest ISO; and partial mangrove mortality PMM). Size classes: I, <40 cm in height (seedlings); II, from 40 to <150 cm in height (juveniles); III, >150 cm in height but with <2.5 cm in diameter at breast height (DBH) (young tree); and IV, >150 cm in height and DBH between 2.5 to >10 cm (adult trees).

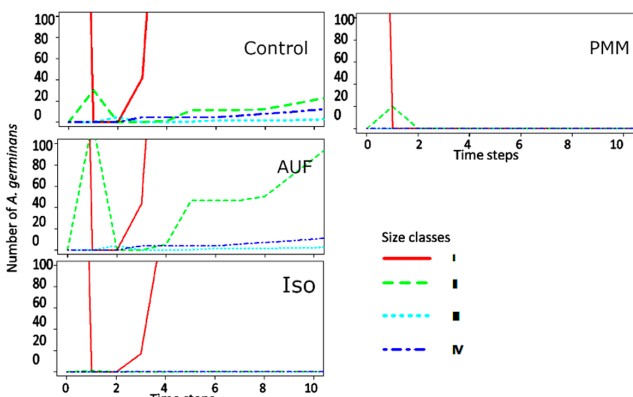

**Figure 8.** Projections based on *A. germinans* at different perturbation levels: Control, apparently unaffected forest (AUF); isolated AUF (ISO) and Partial mangrove mortality (PMM). Size classes: I, <40 cm in height (seedlings); II, from 40 to <150 cm in height (juveniles); III, >150 cm in height but with <2.5 cm in diameter at breast height (DBH) (young tree); and IV, >150 cm in height and DBH between 2.5 to >10 cm (adult trees). Note that the *y*-axis scale was limited to 100 trees to be able to appreciate the dynamic of size classes II to IV, but the projections started with 1000 seedlings in size class I as demonstrated by the red line that rapidly drops from time step 0 to 1, as seedling mortality is large at all the observed areas; in control, AUF and ISO, the number of seedlings recovers and at time steps 3–4, increases to >100 units.

## 4. Discussion

Five species of mangrove trees exist on Mexican mangrove forests [8]; of these, three species dominate (*R. mangle*, *A. germinans*, and *L. racemosa*) on the study area, and give structure to the local forests [14]. At the study site, mean tree density (0.2 trees/m²; 2000 trees/ha) was a relatively high density compared to other locations (220 to 550 trees/ha [34]) but within what is expected for the Gulf of Mexico's mangroves (3360 trees/ha [35]).

Using species and size classes at the study site, we showed that the forest structure differed between levels of perturbation; all three species were absent from the zone of

total mortality (no even seedlings were found), and *A. germinans* was the most abundant species in the partial mortality zone. Several studies have shown that mangrove structure (relative abundance, density by basal area and cover) responds to local environmental factors [36–38]. These forcing factors act on the survival rates of mangrove trees, such as tidal dynamics, water quality, hurricane frequency, and intensity, soil composition, salinity, sedimentation, climatic variability, and direct human effects [39–41].

Of the three species located in the study area, *A. germinans* is the most tolerant to hypersaline conditions and was the most abundant species in the partial mangrove mortality zone [42–44]. The construction of the power plant and embankments to support the power lines may have caused changes in hydrological circulation due to the embankment's location, and this could lead to changes in salinity that may be responsible for the observed mortality [17]. However, despite several restoration efforts, which include opening channels across the embankments, the mangrove has not recovered and continues to deteriorate (Basañez-Muñoz, personal observation).

Characterizing the differences in mangrove structures can help design management strategies as different stages of the trees respond differently to the environment [45,46]. For example, Ref. [47] showed the effect of some environmental variables on recruitment, growth, and survival of *Avicennia marina* (Forsk.); reciprocally, the root system of adult trees of *A. germinans* can modify the soil physicochemical parameters unsuitable for seedlings [48].

We used size-dependent transition probabilities to project the populations of *A. germinans* using modified Leslie matrices [32]. In the models, we used four size categories based on the categories given by [24] but given that trees of different sizes fall on a given category, the actual time elapsed on each time step of the model is difficult to establish. For example, for a seedling to grow to a small tree (transition from size category I to size category II), 20 years might pass [24]. According to the population size structures at all perturbations levels, this transition showed more mortality, particularly for the isolated patch and the partial mortality area. In addition, for the partial mortality area, a lack of trees on the size category III suggests a harsher environment for the trees at this area (Figure 5), which has a strong effect on the population projection (Figure 6). We know that the total mortality zone took about 35 years to reach its actual size and the projection of *A. germinans* in the partial mortality area declined and almost disappeared in approximately two-time steps, which include the passage of seedlings to young trees and the decline of these young trees. This time frame appears to be on a similar temporal scale as the observed loss (35 years). Projections of the control and apparently unaffected sites showed an increase in their populations ($\lambda > 1$). While, at PMM, all size classes showed a rapid decline, at the AUF locations, a relatively high abundance of juveniles and young trees marked the difference between the perturbation levels. Although size distributions alone are not good predictors of population grow [49] the contrasting projections related to the disturbance levels are in accordance with the history of the study area and the forest structural differences. Mangrove species breed repeatedly, and trees from different generations form part of the structure of the forest, yet recruitment alone is not enough to maintain or regenerate a mangrove forest because mortality from size class I to size class II was large at all the observed areas. Climate extremes (cold and warm) and herbivory contribute to seedling mortality [50,51] and the loss at smaller size classes has a strong influence on larger classes [52]. Larger trees suffer senescence and contribute less to recruits, thus conserving larger trees while harvesting relatively young trees was not a good management practice [53]. In fact, our projections show that the presence and survival of young trees enhances the survival of the population. This can happen, for example, by creating microclimates that are adequate for seedling survival [54]. Restoration efforts at the study site have probably failed because attempts to plant seedlings or restore circulation patterns have been made without concern about the forest's structure and the lack of young and adult trees in the areas of complete mortality.

## 5. Conclusions

We have shown that in a relatively small area where three mangrove species are found, different perturbation scenarios coexist and their structure can be differentiated by multivariate techniques such as nMDS, ANOSIM and SIMPER. The mangrove tree *Laguncularia racemosa* (L.) Gaertn. had a low tree and seedling density overall, but at the control and apparently unaffected areas, trees reached large sizes. The partial mortality area differentiated from the rest by having a lower abundance of seedlings. Modified Leslie matrix projections of *Avicennia germinans* (L.) L. showed a declining population at the partial mortality area. Sensitivity analysis showed the importance of seedling recruitment, but also of the survival of young adult trees that might moderate climate extremes and lower herbivory pressure on seedlings. We recommend that future efforts provide seedlings conditions to reach larger sizes, monitor the survival of young adult trees, and examine local abiotic and biotic environmental conditions (for example, soil composition, salinity, nutrients, and herbivory) to create better conditions for the survival of the forest.

**Author Contributions:** Sampling design and collection, data analysis, and writing, A.d.J.B.-M.; Data analysis, writing, and manuscript preparation, A.G.J.-G.; Manuscript review and writing, A.S. All authors have read and agreed to the published version of the manuscript.

**Funding:** This research received no external funding. The APC was funded by Universidad Veracruzana.

**Data Availability Statement:** Publicly available datasets were analyzed in this study. This data can be found here: https://github.com/ajordangarza/Basa-ez-Mu-oz-etal-Mangrove-paper/blob/a2ddfdb25f4ad2df33e3bb019182ef570de2f252/Basa%C3%B1ez-Mu%C3%B1oz%20etal%20Magrove%20forest%20(1).xlsx (accessed on 26 July 2021).

**Acknowledgments:** We would like to thank the Universidad Veracruzana and particularly, the Facultad de Ciencias Biológicas y Agropecuarias for supporting the realization of this project. Additionally, we would like to acknowledge the "Comisión Federal de Electricidad" for granting us access to the study site.

**Conflicts of Interest:** The authors declare no conflict of interest.

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
