# Peer review of "Forest Structure and Projections of Avicennia germinans (L.) L. at Three Levels of Perturbation in a Southwestern Gulf of Mexico Mangrove"

_forests, doi:10.3390/f12080989_

Round 1
Reviewer 1 Report
Dear Authors,
I revised your manuscript entitled "Forest structure and projections of Avicennia germinans at three levels of perturbation in a southwestern Gulf of Mexico mangrove". I think it is very interesting and can help to broaden understand the dynamics of some mangrove communities under pressure. Although it is well structured, it needs some important improvements. In particular, the "Introduction" section needs to be expanded a little in the initial part, introducing some more details about mangrove communities in other parts of the world.
Line 39. Since you already report reference n. 12 in next sentence, please, change it with another appropriate, such as the following: Cano-Ortiz, A.; Musarella, C.M.; Piñar, J.C.; Pinto Gomes, C.J.; del Río González, S.; Cano, E. Diversity and conservation status of mangrove communities in two areas of Mesocaribea biogeographic region. Curr. Sci. 2018, 115, 534–540. doi: 10.18520/cs/v115/i3/534-540
Pay close attention to correctly report the scientific names in the title, main text, figures, tables and their captions, as indicated by me in the attached PDF.
Although the "Conclusions" section is not mandatory, it can be added to the end of your manuscript to better summarize your findings and focus the conclusions of your work from the "Discussion" section.
Finally, English needs to be improved, as some sentences are too complex or unclear.
Other notes are reported in the attached PDF.
After all these improvements the manuscript, in my opinion, could continue in its publication process.
Best wishes.

Reviewer 2 Report
Dear editors, dear authors,
The manuscript presents an interesting topic on characterising mangrove community structure based on the perturbation levels due to (mainly) anthropogenic activities. The manuscript aimed to understand how the mangroves respond to the disturbances based on several levels by taking the Southwestern Gulf of Mexico as the case study. The data based on the transects and 10x10m plots have been collected. The study showed an interesting fact worth notice that A.germinans is capable of surviving in three out of four perturbation levels.
The manuscript successfully described the character differences in mangrove structures based on the abundant collected field data, which is much appreciated. The analysis is based on well-established methods and statistics tools. It, therefore, merits to be published if the methodological considerations and background can be explained further in the manuscript. The material and methods section can be elaborated further to describe more on the data collection technique (how and why did you choose the technique) and elaborate more on the statistical analysis by providing details on the consideration of choosing them.
Despite the well-described results on the data, the manuscript should consider adding more discussion on how the recruitment and abundance of juvenile and young trees are the reasons for the stability of the mangrove forest. The discussion section in paragraph five shows the hypothesis and reasons for the decline, but did not yet answer the reasons for the stability; please add the argumentation.
Ironically, the conclusion section is missing in this manuscript, a mandatory part of the paper structure. Thus, as a reader, we don’t know what the take-home message of the manuscript is and what is new on the finding.
Minor Issues are provided in the attachment.
Best regards,

Reviewer 3 Report
Line 140: please add the appropriate brackests in the Leslie model: Matrix applied to a vector.
In Tables 1 and 2 the standard deviation of the mean of most variables (mainly densities) give negative confidence intervals. You should use Box Plots instead.
Fig. 6. Explain, why the numbers of size I class trees exceed the scale of the figure or use a logarithmic scale.
Remark: the use of the Leslie projection model is quite adequate!
Round 2
Reviewer 1 Report
Dear Authors,
Many thanks for the trust.
Best wishes!
Reviewer 2 Report
Dear authors,
Thank you for the response and revised paper.
The responses have been addressed, and the paper has been improved.
An addition in the description of perturbation zones categorization and how the perturbation zones were defined will give more understanding to the reader.
Moreover, slight corrections should be made in the editing stage.
Best regards